# Prevalence of symptomatic dry eye disease with associated risk factors among medical students at Chiang Mai University due to increased screen time and stress during COVID-19 pandemic

Chulaluck Tangmonkongvoragul[1]*, Susama Chokesuwattanaskul[1], Chetupon Khankaeo[2], Ruethairat Punyasevee[2], Lapat Nakkara[2], Suttipat Moolsan[2], Onpreeya Unruan[2]

1 Department of Ophthalmology, Faculty of Medicine, Chiang Mai University, Chiang Mai, Thailand,
2 Faculty of Medicine, Chiang Mai University, Chiang Mai, Thailand

* chulaluck.t@cmu.ac.th

## Abstract

Dry eye disease (DED) is one of the most common ophthalmological disorders, resulting from several systemic and ocular etiologies including meibomian gland dysfunction (MGD). During the COVID-19 pandemic, medical students are among the high-risk group for DED, mainly due to the increasing use of a visual display terminal (VDT) for online lectures and psychological stress from encountering several changes. Our study aimed to explore the prevalence of DED using the symptom-based definition and potential risk factors in medical students. This is a prospective cross-sectional study that included medical students at Chiang Mai University between November 2020 and January 2021. All participants were assessed using the Ocular Surface Disease Index (OSDI) questionnaire, the Thai version of the 10-Item Perceived Stress Scale-10 (T-PSS-10), the LipiView® II interferometer, and an interview for other possible risk factors. Overall, 528 participants were included in the study; half of the participants were female. The prevalence of DED was 70.8%. In the univariate analysis, female sex, contact lens wear, and T-PSS-10 stress scores were significantly higher in the DED group (*P* = 0.002, 0.002, and <0.001, respectively). Moreover, participants with severe DED were likely to have higher meibomian gland tortuosity but not statistically significant. In the multivariate analysis, contact lens use and T-PSS-10 score were significant risk factors associated with the severity of DED. In conclusions, the prevalence of DED in medical students was as high as 70.8%. Contact lens use and psychological stress evaluated using the T-PSS-10 questionnaire had a significant correlation with a risk of DED. Female gender and duration of VDT use were also associated. Most of the risk factors were modifiable and may be used as initial management in patients with DED.

**Data Availability Statement:** Due to the potentially sensitive information, the datasets used and/or analyzed in this study are available on reasonable request. The contact information is: Department of Ophthalmology, Faculty of Medicine, Chiang Mai University, 110 Intawarorot Road, Suthep, Muang, Chiang Mail, 50200, Thailand, Telephone number +66-53-935512, Email: ekasit_ka@cmu.ac.th.

**Funding:** This research was funded by Faculty of medicine, Chiang Mai University, Chiang Mai, Thailand (Grant number: 043/2564). The funder has no role in the study design, data collection and analysis, decision to publish, or preparation of the manuscript.

**Competing interests:** The authors have declared that no competing interests exist.

## Introduction

In 2017, the Dry Eye Workshop II (DEWS II) organized by the Tear Film & Ocular Surface Society (TFOS) defined dry eye as a multifactorial disease of the ocular surface characterized by a loss of the tear film homeostasis, accompanied by ocular symptoms. Additionally, the tear film instability and hyperosmolarity, ocular surface inflammation and damage, and neurosensory abnormalities play critical roles [1]. The dry eye symptoms (DES) may include ocular dryness, discomfort, pain, grittiness, blurry vision, redness, foreign-body sensation, and visual disturbance, which could significantly disturb daily life activities such as reading, driving, and using visual display terminals (VDT) [2–4].

The reported prevalence of dry eye has been varied among studies due to the inconsistent definition and diagnostic criteria. The meta-analysis yields prevalence of dry eye ranging from 5% to 50% with symptom-based diagnosis and as high as 75% with any positive ocular signs [5]. The prevalence of dry eye disease (DED) in Asians is higher than in Caucasians [5, 6]. In the adult Thai population, the prevalence of DED is reported to be 34% by questionnaire [6].

The identification of DED subtypes, either aqueous deficient or evaporative, is essential for classification and management. However, these subtypes are more considered as a spectrum of disease rather than distinct pathophysiological entities. Evaporative DED is more common, in which meibomian gland dysfunction (MGD) is the most common etiology. The international workshop defined MGD as a chronic, diffuse abnormality of the meibomian glands, usually characterized by an obstruction of terminal duct and/or changes in the secretion of the glands qualitatively/quantitatively. The condition may lead to a tear film alteration, symptoms of eye irritation and inflammation, and ocular surface disease [7]. In a previous study, 65–86% of dry eye patients had MGD [8–10].

Several studies have reported the prevalence and risk factors of DED in university students [11–13]. However, there is limited data regarding the characteristics of DED in medical students [14–17]. Moreover, during the COVID-19 pandemic, medical students are at high-risk for developing dry eye symptoms due to increasing use of VDT for online lectures and psychological stress from encountering several changes. Our study aimed to explore the prevalence and potential risk factors of DED among medical students at Chiang Mai University, Thailand.

## Materials and methods

This prospective cross-sectional descriptive study included 528 medical students attending Chiang Mai University, Thailand, between November 2020 and January 2021 with informed consent. The study was conducted in accordance with the tenets of the declaration of Helsinki, and the protocol was approved by the Faculty Ethics Committee. Only medical students who completed the questionnaire and the LipiView® II interferometer examination were included in the analysis. All participants can refuse to be in the study at all, or to stop participating at any time of the study.

A semi-structured questionnaire was designed to assess the dry eye symptoms and their potential risk factors among the medical students during the COVID-19 pandemic. The survey questionnaire had four sections: demographic data with pre-existing medical conditions; risk factors for DED including personal habits; psychological stress; and a dry eye questionnaire using the Ocular Surface Disease Index (OSDI). After completing all questionnaires in Google forms, all participants were examined with the LipiView® ocular surface interferometer.

Participants who had a history of ocular surgery or trauma within 3 months, or ocular diseases such as ocular infection, allergy, autoimmune disease, and those using punctal plug or

topical ocular medications other than artificial tears were excluded. Participants who used artificial tears were instructed to stop the use 6 hours before LipiView® II examination.

## Ocular Surface Disease Index (OSDI)

The OSDI (Allergan, Inc, Irvine, California) questionnaire is comprised of 12 questions assessing three domains of the ocular surface diseases in dry eyes as follows: 5 questions regarding the dry eye symptoms related to chronic dry eye disease; 4 regarding the limited visual performance related to dry eyes; and 3 regarding the severity of the symptoms in specific conditions during the previous week. Overall scores (range from 0 to100) were calculated and categorized into 3 groups: normal (score 0–12); mild symptoms (score 13–22); moderate symptoms (score 23–32); and severe symptoms (score 33–100) [18].

## Psychological stress

The Thai version of the 10-Item Perceived Stress Scale-10 (T-PSS-10) was used to evaluate the psychological stress related to dry eye symptoms. Of 10 questions, 4 positive questions and 6 negative questions were used to quantify the individual's perceived psychological stress. The scores range from 0 to 40, with higher scores associated with increased perceived psychological stress [19].

## LipiView® II ocular surface interferometer

The LipiView® II ocular surface interferometer (TearScience Inc., Morrisville, NC, USA) was used to measure lipid layer thickness (LLT), meibomian gland dropout, meibomian gland dilatation (tortuosity) and blinking pattern. During the test, participants were instructed to maintain fixation on the internal target. All participants were allowed to blink naturally during the image captured, which is typically evaluated for 20 seconds [20].

A single experienced observer (CT) subjectively evaluated the meibomian gland dropout in both upper and lower eyelids using a validated Meibograde grading scheme, with a 4-point scale from 0 to 3 in which grade 0 is 0–25% meibomian gland loss; grade 1, 26–50% loss; grade 2, 51–75% loss; and grade 3, more than 75% loss [21]. The meibomian gland loss was calculated with reference to the equivalent meibomian gland area in healthy individuals. Meibomian gland tortuosity for each eyelid was graded using the 5-point Halleran scale: grade 0, no tortuosity; grade 1, less than 25% tortuosity; grade 2, 26–50% tortuosity; grade 3, 51–74% tortuosity; and grade 4, more than 75% tortuosity [22]. Each eyelid was blinded for evaluation and the upper and lower lids were separately evaluated. The cut-off value for LLT is 60 nm, where the LLT ≤60 nm indicates a chance of MGD with 90% specificity [23]. For the blinking pattern, the incomplete blinking ratio was calculated by the number of incomplete blinks divided by the total blinks [20].

## Prevalence of meibomian gland dysfunction (MGD)

Using the validated sensitivity and specificity of the Meibograde as a diagnostic parameter for MGD, as described by Adil *et al.*, a cut-off value of average Meibograde of 0.5 yielded a sensitivity and specificity of 96.7% and 85%, respectively, while a cut-off value of average Meibograde of 0.75 yielded a sensitivity and specificity of 87.9% and 100%, respectively [21].

## Statistical analysis

SPSS software for Windows version 25.0 (Armonk, NY: IBM Corp.) was used for the statistical analyses. Data were tested of normal distribution using the Kolmogorov-Smirnov test with the

cut-point $P$-value of 0.05. The prevalence was presented with a mean and 95% confidence interval (CI). For the univariate analysis, the Kruskal-Wallis test and Chi-square test were used for categorical variables, the Mann-Whitney U test and ANOVA test for quantitative variables. For the multivariate analysis, the binary logistic regression was performed. A $P$-value <0.05 was considered as statistically significant.

## Results

### Prevalence of dry eye symptoms (DES)

A total of 528 medical students completed all questionnaires and the meibomian gland evaluation using the LipiView® II interferometer. Of those, 252 (47.4%) were male, and 276 (52.3%) were female. The mean (range) age was 20.48 (17–31) years. For the refractive errors, 78.6% (415/528) of the medical students had myopia and 8.71% (46/528) had hyperopia. The use of spectacles (392/528, 74.24%) was more common than contact lenses (69/528, 13.07%). The prevalence of DED based on symptoms (OSDI >12) was 70.8%.

### Risk factors associated with dry eye symptoms (DES)

Univariate analysis showed that female sex ($P$ = 0.002), contact lens wear ($P$ = 0.002), prolonged hours of contact lens wear ($P$ = 0.004), higher frequency of artificial tears used per day ($P$ = 0.001) and higher score of T-PSS-10 ($P$ <0.001) were associated with increased risk of dry eye symptoms (DES). However, the history of refractive surgery, duration of VDT use, and hours of reading paperwork were not significantly different between those with and without DES. The results from LipiView® II, including incomplete blink to total blink ratio, Meibograde and meibomian gland tortuosity scores, were higher in the dry eye group but not statistically significant. (**Table 1**)

The prevalence of mild, moderate, and severe dry eyes in medical students based on the OSDI score were 24.2%, 18.8% and 27.8%, respectively. **Table 2** shows the results of the univariate analysis of potential risk factors for DES according to the severity of dry eyes. Female sex ($P$ = 0.005), contact lens wear ($P$ = 0.005), prolonged hours of contact lens wear ($P$ <0.001), higher frequency use of artificial tears per day ($P$ = 0.003), longer duration of VDT use per day ($P$ = 0.033) and higher score of T-PSS-10 (P <0.001) were also associated with increased severity of dry eyes. According to LipiView® II, only the meibomian gland tortuosity increased with higher severity of dry eyes, but was not statistically significant. Results of Meibography using Meibograde grading scheme were shown in **Fig 1** and Meibomian gland tortuosity in **Fig 2**.

The results of regression analysis are summarized in **Table 3**. Higher T-PSS-10 score (OR, 1.113; 95% CI, 1.074–1.154; P <0.001) and contact lens wear (OR, 0.287; 95% CI, 0.134–0.615; P = 0.001) were significant risk factors associated with the severity of DES.

### Prevalence of meibomian gland dysfunction (MGD)

Using an average Meibograde of 0.75 as a cut-off value, the prevalence of total MGD in medical students was 60.98% (322 in 528) with 100% specificity of MGD diagnosis [21]. Of these, the prevalence of asymptomatic MGD (OSDI score 0–12 with Meibograde cut off ≥ 0.75) was 17.61% (93 in 528) and symptomatic MGD (OSDI score > 12 with Meibograde cut off ≥ 0.75) was 43.37% (229 in 528).

**Table 1. Univariate analysis of potential risk factors for DES according to the presence and absence of dry eye symptoms.**

| Parameters | No symptoms of dry eyes (OSDI score 0–12) N = 154 | Presence of symptoms of dry eyes (OSDI score >12) N = 374 | P-value |
|---|---|---|---|
| OSDI score | 5.95±3.37 | 30.05±13.56 | < 0.001[†] |
| Age (years) | 20.73±1.60 | 20.38±1.68 | 0.130[†] |
| Sex (M:F) (n) | 90:64 | 162:212 | 0.002[*] |
| Myopia (%) | 72.7 | 81.0 | 0.035[*] |
| Glasses wear (%) | 68.2 | 76.7 | 0.041[*] |
| Contact lens wear (%) | 5.8 | 16.0 | 0.002[*] |
| • Daily contact lens | 4.5 | 6.4 | |
| • Monthly contact lens | 1.9 | 10.2 | |
| Hours of contact lens wear (hours/day) | 0.60±2.30 | 1.68±4.04 | 0.004[†] |
| Frequency of artificial tears used per day (times/day) | 0.12±0.71 | 0.28±0.84 | 0.001[†] |
| History of refractive surgery (%) | 0.6 | 1.6 | 0.679[*] |
| Duration of VDT use per day (hours) | 9.55±3.13 | 9.88±3.12 | 0.252[†] |
| Hours of paperwork per day (hours) | 0.95±1.31 | 0.88±1.04 | 0.997[†] |
| Stress score (T-PSS-10) | 12.84±5.72 | 16.27±5.69 | < 0.001[†] |
| Average incomplete blink-Total blink ratio of both eyes | 0.62±0.32 | 0.66±0.29 | 0.285 |
| Average lipid thickness of both eyes (nm) | 61.23±21.20 | 62.72±19.59 | 0.302 |
| **Meibograde** | | | |
| Average Meibograde of all 4 eyelids (0–3 scale) | 0.84±0.59 | 0.87±0.63 | 0.665[†] |
| **Meibomian Gland Tortuosity** | | | |
| Average Meibomian gland tortuosity of all 4 eyelids (0–4 scale) | 1.57±0.66 | 1.67±1.15 | 0.561[†] |

All data are expressed as mean ± SD, or percentage, as appropriate.

[*]P-value was calculated using the Kruskal-Wallis test.

[†]P-value was calculated using the Mann-Whitney U test.

## Discussion

Dry eye disease is the most common presenting ocular surface disease in ophthalmic practice. Though the nature of DED is complex, the key pathophysiology is basically the disruption of tear film homeostasis [1]. However, the signs and symptoms of DED are sometimes poorly correlated. Consequently, no single gold-standard is accepted as a diagnostic marker for DED. For the clinical diagnosis of DED, TOFS DEWS II recommend the use of DED questionnaires to determine the subjective severity of the symptoms and their sequelae on quality-of-life. Additionally, the presence of at least one clinical sign of abnormal tear film homeostasis is required for the diagnosis. Positive ocular signs include decreased tear break-up time, tear film hyperosmolarity, and ocular surface staining [24]. However, in our study, the diagnosis of DED was made solely on the presence of DES, as defined by an OSDI score >12. This symptom-based definition has also been widely accepted in clinical practice and research, especially in the large population-based studies [4, 15, 25–27].

Previous studies revealed that the DED, using the Schirmer test for diagnosis, was commonly found in medical students [14, 16]. The study of Yang I *et al.* in 2019 in Brazilian medical students, which the DED was diagnosed with OSDI score, keratography, ocular surface staining and the Schirmer test, the prevalence of severe dry eyes (OSDI score >33) was 12.6% compared with 27.8% in our study. They also found that the duration of VDT use and contact

**Table 2. Univariate analysis of potential risk factors according to the severity of dry eye symptoms.**

| Parameters | Normal (OSDI 0–12) n = 154 | Mild dry eyes (OSDI 13–22) n = 128 | Moderate dry eyes (OSDI 23–32) n = 99 | Severe dry eyes (OSDI >33) n = 147 | P-value |
|---|---|---|---|---|---|
| OSDI score | 5.95±3.37 | 16.55±2.53 | 26.92±2.99 | 43.92±9.92 | < 0.001[†] |
| Age (year) | 20.73±1.60 | 20.33±1.74 | 20.34±1.62 | 20.45±1.68 | 0.890* |
| Sex (M:F) | 90:64 | 63:65 | 40:59 | 59:88 | 0.005* |
| Myopia (%) | 72.7 | 74.2 | 85.9 | 83.7 | 0.019* |
| Glasses wear (%) | 68.2 | 70.3 | 84.8 | 76.9 | 0.016* |
| Contact lens wear (%) | 5.8 | 13.3 | 14.1 | 19.7 | 0.005* |
| • Daily contact lens | 4.5 | 2.3 | 7.1 | 9.5 | |
| • Monthly contact lens | 1.9 | 10.9 | 7.1 | 11.1 | |
| Hours of contact lens wear (hours/day) | 0.60±2.30 | 1.11±3.20 | 1.41±3.68 | 2.35±4.80 | 0.000[†] |
| Frequency of artificial tears used per day (times/day) | 0.12±0.71 | 0.15±0.58 | 0.20±0.74 | 0.44±1.05 | 0.003[†] |
| History of refractive surgery (%) | 0.6 | 0.8 | 3.0 | 1.4 | 0.388* |
| Duration of VDT use per day (hr) | 9.55±3.12 | 9.27±3.00 | 10.21±3.07 | 10.19±3.20 | 0.033[†] |
| Hours of paperwork per day (hr) | 0.95±1.31 | 0.92±1.11 | 0.80±1.12 | 0.89±0.90 | 0.738[†] |
| Stress score (T-PSS-10) | 12.84±5.72 | 15.73±6.20 | 15.81±5.28 | 17.05±5.43 | < 0.001[†] |
| Average incomplete blink-Total blink ratio of both eyes | 0.62±0.32 | 0.65±0.30 | 0.69±0.28 | 0.65±0.28 | 0.361[†] |
| Average lipid thickness of both eyes (nm) | 61.23±21.20 | 64.17±19.71 | 63.82±19.38 | 60.72±19.59 | 0.388[†] |
| **Meibograde** | | | | | |
| Average Meibograde of all 4 eyelids (0–3 scale) | 0.84±0.59 | 0.82±0.64 | 0.91±0.66 | 0.90±0.60 | 0.593[†] |
| **Meibomian Gland Tortuosity** | | | | | |
| Average Meibomian gland tortuosity of all 4 eyelids (0–4 scale) | 1.57±0.66 | 1.59±0.72 | 1.65±0.84 | 1.74±1.55 | 0.474[†] |

All data are expressed as mean ± SD, or percentage, as appropriate.

*P-value was calculated using the Kruskal-Wallis test.

[†]P-value was calculated using ANOVA.

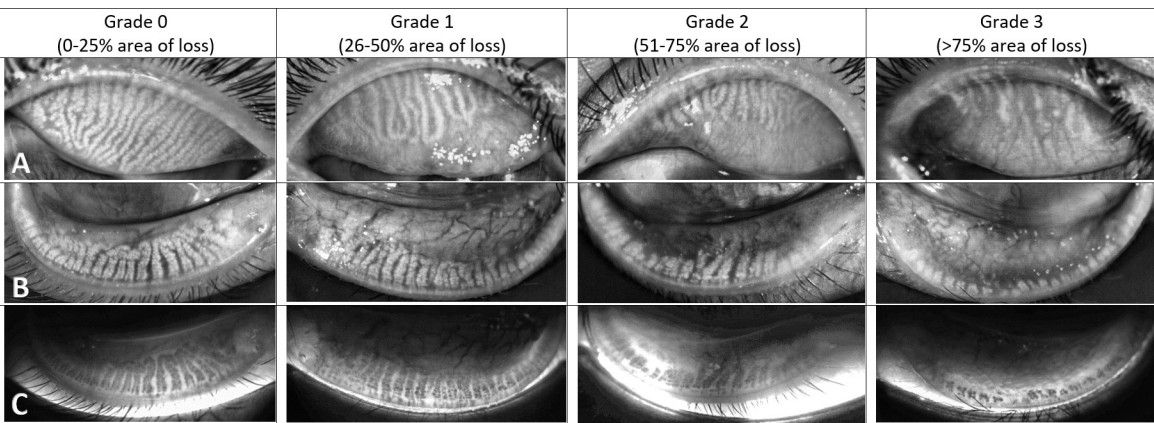

**Fig 1. The Meibograde grading system: subjective grading of meibomian gland loss.** Row A: Dynamic illumination mode (Reflect infrared) of upper lids; Row B: Dynamic illumination mode (Reflect infrared) of lower lids; Row C: Adaptive transillumination mode (Trans infrared) of lower lids.

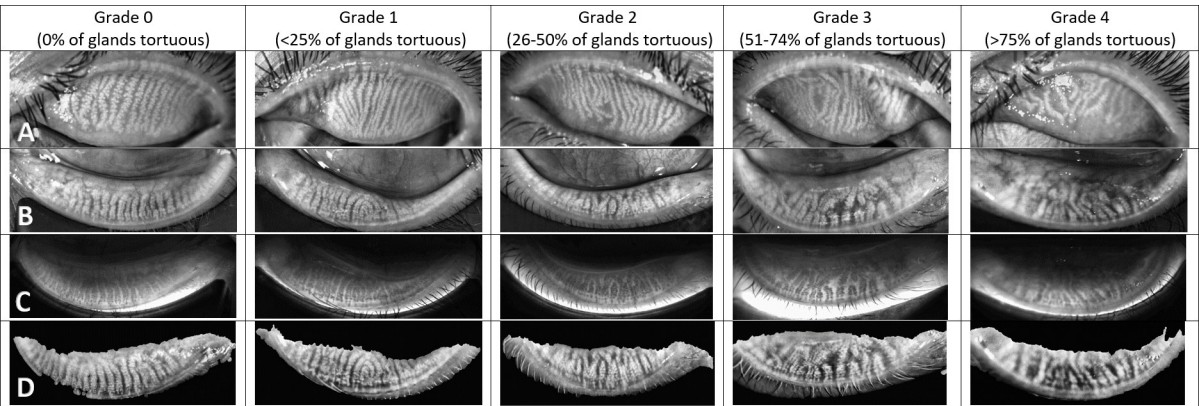

| Grade 0 (0% of glands tortuous) | Grade 1 (<25% of glands tortuous) | Grade 2 (26-50% of glands tortuous) | Grade 3 (51-74% of glands tortuous) | Grade 4 (>75% of glands tortuous) |

**Fig 2. The Meibomian gland tortuosity grading system.** Row A: Dynamic illumination mode (Reflect infrared) of upper lids; Row B: Dynamic illumination mode (Reflect infrared) of lower lids; Row C: Adaptive transillumination mode (Trans infrared) of lower lids; Row D: Dual mode (combine dynamic illumination and adaptive transillumination) of lower lids.

lens wear increased the risk of DED [17]. Hyon *et al.* demonstrated that stress, female sex, contact lens wear, and duration of using VDT were significant risk factors for DED. The prevalence of DED based on the symptoms in Korean medical students was 27.1% This study evaluated DED in medical students before the COVID-19 pandemics and revealed that DED may have association with psychological stress (using the Perceived Stress Scale 4 (PSS-4) questionnaire) [15]. Additionally, the PSS scores in our study tended to be higher than the study by Hyon *et al.*, though the direct comparison was not allowed due to the different versions used. Therefore, a higher prevalence of symptomatic DED in medical students (70.8%) in our study may be explained by more stressful situation during the COVID-19 pandemic among medical students. The study of engineering students in Tamilnadu, South India also showed a high prevalence of symptom-based DED, 64.1% [28]. This study was similarly conducted during the ongoing COVID-19 pandemic. Higher prevalence of DED was presumably due to increased screen time use and stress.

Our study revealed that female sex was likely to be associated with an increased risk of DED, similar to the previous studies [4, 6, 25, 27, 29–33]. Hyon *et al.* revealed an association between female gender and the development of DED in medical students (P = 0.026) [15]. In addition, females reported higher rates of dry eye symptom scores impacting their quality of life, more negative side effects from the treatments, and longer time for improvement than males [30, 33]. The higher prevalence of DED in females may be partially explained by the balance of sex hormones, particularly androgen that affects the synthesis and interaction of tear film components [6, 25, 31]. Moreover, there are anatomic differences of the meibomian

**Table 3. Regression analysis for factors associated with increasing severity of DES.**

| Variables | Adjusted for sex, contact lens wear, duration of VDT use and stress score | |
| --- | --- | --- |
| | **Odds ratio (95%CI)** | **_P_-value** |
| Female | 0.672 (0.449–1.004) | 0.052 |
| Contact lens wear | 0.287 (0.134–0.615) | 0.001 |
| Duration of VDT use per day (hr) | 1.027 (0.961–1.097) | 0.431 |
| T-PSS-10 (score) | 1.113 (1.074–1.154) | < 0.001 |

Abbreviation: CI = confident interval.

glands lacrimal apparatus. However, the definite pathophysiology underlying the increased risk of DED in females remains unclear.

Contact lens wear was also one of the associated risks of DED. Possible mechanisms include increased tear evaporation, ocular surface changes, reduced density of the goblet cells, altered mucin production, and meibomian gland dropouts, eventually leading to the disturbance of the tear film homeostasis [34]. Many studies show that contact lens wear along with VDT use significantly increased the risk of DED due to disrupting the tear film stability [4, 27, 35].

This study showed that a longer duration of VDT use was possibly associated with an increased risk of DED, consistent with prior studies [4, 32]. In 2018, Iqbal *et al*. reported that as high as 68% of the students who used the devices for >13 hours per day developed the DED and 28% of students with only >3 hours of screen time had symptoms of dryness [36]. Prolonged VDT use may interfere with the tear film instability due to decreased blinking rates and increased tear evaporation [4, 37, 38]. COVID-19 pandemic may impact the development of DES among medical students due to increased digital screen time for online lectures and stress from the spreading of COVID-19. Health promotion with the "20-20-20" rule, which recommends that every 20 minutes, an individual should take a 20-second break and focus their eyes on something at least 20 feet away. and limited screen time may be very helpful.

T-PSS-10 score had a significant association with DED in both the univariate and multivariate analysis. Our study demonstrated the association between psychological stress and DED. Hyon *et al*. used the stress VAS and PSS-4 and found a similar association in medical students and paramedical workers [15, 26]. Moreover, previous studies show an association between DED and several psychiatric conditions, including depression and post-traumatic stress disorder [39, 40]. For this possible association of psychological stress and DED, medical students are generally living under pressure from high expectations and are at high-risk for developing DED. Further studies, with larger populations, are necessary to evaluate the pathophysiology underlying the association between the DED and stress.

Meibomian gland features and tear meniscus assessment can be used to classify the predominant DED subtypes and severity to guide the proper management. Meibography is an objective test for accurately diagnosing MGD and its severity [41]. The gland area dropout is a quantification grading system widely used to diagnose MGD, either alone or in combination with other tests. When using the cut-off value of 0.5 for average Meibograde, the sensitivity and specificity were 96.7% and 85%, respectively. With a higher average Meibograde of 0.75 as a cut-off value, the sensitivity and specificity shifted to 87.9% and 100%, respectively [21]. Previous studies reported that 65–86% of dry eye patients have MGD [8–10, 41]. Therefore, since the prevalence of DED was 70.8% in our study, the prevalence of MGD should be approximately 46–60.8%. Consistently, the actual prevalence of MGD using Meibograde criteria was 60.98%. Notably, most MGD patients were symptomatic.

There were limitations to this study. Firstly, the data was collected from medical students in a single university, which may not represent all medical students in Thailand. Secondly, no clinical evaluations for the dry eye signs were performed. Though several studies applied the symptom-based diagnosis of DED, the clinical examination is still mandatory for the diagnosis of DED in some cases [24]. Moreover, the prevalence of DED may be varied from the different diagnostic methods. Thirdly, the data was collected during the COVID-19 pandemic, in which medical students had an increase in both digital screen time for online lectures and stress. The prevalence of DED in a normal situation may be different. Further study should be performed in a non-pandemic situation and include medical students from various universities for comparison.

## Conclusions

This study revealed a high prevalence of symptomatic DED in medical students (70.8%). Contact lens wear and psychological stress evaluated using the T-PSS-10 questionnaire had a significant correlation with the risk of DED. Female gender, prolonged hours of contact lens wear, higher frequency use of artificial tears per day, and prolonged duration of VDT use was also associated with increased severity and risk of DED. Meibomian gland tortuosity was increased with higher severity of dry eyes but not statistically significant. Using an average Meibograde of 0.75 as a cut-off value, the prevalence of MGD in medical students was 60.98% and was mostly symptomatic.

## Supporting information

**S1 File. Questionnaire for Demographic data and risk factors for DED_EN version.**
(PDF)

**S2 File. Questionnaire for Demographic data and risk factors for DED_TH version.**
(PDF)

**S3 File. Perceived Stress Scale-10 (PSS-10).**
(PDF)

**S4 File. Thai Perceived Stress Scale-10 (T-PSS-10).**
(PDF)

**S5 File. OSDI_EN version.**
(PDF)

**S6 File. OSDI_TH version.**
(PDF)

## Acknowledgments

We would like to thank Ms. Barbara Metzler, a director of the Chiang Mai University English Language Team, for help with manuscript editing.

## Author Contributions

**Conceptualization:** Chulaluck Tangmonkongvoragul, Susama Chokesuwattanaskul.

**Data curation:** Chulaluck Tangmonkongvoragul, Susama Chokesuwattanaskul, Chetupon Khankaeo, Ruethairat Punyasevee, Lapat Nakkara, Suttipat Moolsan, Onpreeya Unruan.

**Formal analysis:** Chulaluck Tangmonkongvoragul, Susama Chokesuwattanaskul.

**Funding acquisition:** Chulaluck Tangmonkongvoragul.

**Investigation:** Chulaluck Tangmonkongvoragul, Susama Chokesuwattanaskul, Chetupon Khankaeo, Ruethairat Punyasevee, Lapat Nakkara, Suttipat Moolsan, Onpreeya Unruan.

**Methodology:** Chulaluck Tangmonkongvoragul.

**Project administration:** Chulaluck Tangmonkongvoragul.

**Resources:** Chulaluck Tangmonkongvoragul.

**Supervision:** Chulaluck Tangmonkongvoragul.

**Validation:** Chulaluck Tangmonkongvoragul, Susama Chokesuwattanaskul.

**Visualization:** Chulaluck Tangmonkongvoragul, Susama Chokesuwattanaskul.

**Writing – original draft:** Susama Chokesuwattanaskul.

**Writing – review & editing:** Chulaluck Tangmonkongvoragul, Susama Chokesuwattanaskul.

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
