## [Decision Letter · Decision Letter 0]

25 Jun 2021

PONE-D-21-18381

Prevalence of symptomatic Dry eye disease with associated risk factors among medical students at Chiang Mai University during COVID-19 pandemic

PLOS ONE

Dear Dr. Tangmonkongvoragul,

Thank you for submitting your manuscript to PLOS ONE. After careful consideration, we feel that it has merit but does not fully meet PLOS ONE’s publication criteria as it currently stands. Therefore, we invite you to submit a revised version of the manuscript that addresses the points raised during the review process.

We look forward to receiving your revised manuscript.

Kind regards,

Michael Mimouni

Academic Editor

PLOS ONE

Journal Requirements:

"This research was funded by Faculty of medicine, Chiang Mai University, Chiang Mai, Thailand

(Grant number: 043/2564). The funder has no role in the study design, data collection and analysis, decision

to publish, or preparation of the manuscript."

"This research was funded by Faculty of medicine, Chiang Mai University, Chiang Mai, Thailand (Grant number: 043/2564). The funder has no role in the study design, data collection and analysis, decision to publish, or preparation of the manuscript. "

4. We note that Figures 1 and 2 in your submission contain copyrighted images. All PLOS content is published under the Creative Commons Attribution License (CC BY 4.0), which means that the manuscript, images, and Supporting Information files will be freely available online, and any third party is permitted to access, download, copy, distribute, and use these materials in any way, even commercially, with proper attribution. For more information, see our copyright guidelines: http://journals.plos.org/plosone/s/licenses-and-copyright.

a. You may seek permission from the original copyright holder of Figures 1 and 2 to publish the content specifically under the CC BY 4.0 license. 

Reviewers' comments:

Reviewer's Responses to Questions

**Comments to the Author**

1. Is the manuscript technically sound, and do the data support the conclusions?

Reviewer #1: Partly

Reviewer #2: Yes

Reviewer #3: Partly

2. Has the statistical analysis been performed appropriately and rigorously? 

Reviewer #1: Yes

Reviewer #2: Yes

Reviewer #3: N/A

3. Have the authors made all data underlying the findings in their manuscript fully available?

Reviewer #1: Yes

Reviewer #2: Yes

Reviewer #3: Yes

4. Is the manuscript presented in an intelligible fashion and written in standard English?

Reviewer #1: Yes

Reviewer #2: Yes

Reviewer #3: Yes

5. Review Comments to the Author

Reviewer #1: Chulaluck et al. describe a prospective cross-sectional study aimed to explore the prevalence of symptomatic Dry eye disease (DED) with associated risk factors among medical students at Chiang Mai University during covid-19 pandemic

Comments:

1. Abstract: The study did not include any clinical evaluation of the participants; therefore some of them could potentially have other ocular conditions that may have caused dry eye symptoms.

2. The use of a surgical face mask – which is known to increase dry eye symptoms especially during the COVID-19 pandemic, was not part of the risk factors taken in consideration.

3. The information collected during the study only represented the participants' condition at the day of their examination.

4. Methods section: was not elaborate enough. Exclusion and inclusion criteria are not detailed enough.

5. The LipiView test is designed to measure the thickness of the lipid layer, but recent studies showed that It doesn’t seem to correlate too well with people’s slit lamp exams or symptoms.

6. The study only included medical students from Chiang Mai University, which may not represent all medical students in Thailand.

Reviewer #2: We would like to thank the authors for their work on the Prevalence of symptomatic Dry eye disease with associated risk factors among medical students at Chiang Mai University during COVID-19 pandemic. We think it is a well written paper.

We still would like to give some advice on correction:

1.) Please correct the punctuations. There are lots of mistakes regarding the commas options

Introduction:

2.) Please change “The prevalence of dry eye disease (DED) in Asians is higher than Caucasians.” To “The prevalence of dry eye disease (DED) in Asians is higher than in Caucasians. “ and name a reference

3.) “The international workshop defined MGD as a chronic, diffuse abnormality of the meibomian glands, usually characterized by an obstruction of terminal duct and/or changes in the secretion of the glands qualitatively/quantitatively. Please name a reference

4.) The survey questionnaire had four sections: demographic data with pre-existing medical conditions; risk factors for DED including personal habits; psychological stress; and a dry eye questionnaire using the Ocular Surface Disease Index (OSDI), please attach the questionnaire

5.) Please change “Participants who used artificial tears were instructed to stop using for 6 hours before LipiView® II examination. “ to “Participants who used artificial tears were instructed to stop the use 6 hours before LipiView® II examination.”

LipiView® II Ocular Surface interferometer:

6.) “For the blinking pattern, the incomplete blinking ratio was calculated by the number of incomplete blinks divided by the total blinks.” Please refer to the attached picture

Results:

7.) 392/528 is 74,24% not 85%

8.) 69/528 is 13% not 15%

Table 1:

9.)

10.) Please change “average” to “average”

Discussion:

11.) “Hyon et al.“ Please also name the reference here

12.) “Health promotion with the “20-20-20” rule and limited screen time may be very helpful.“ Please explain this rule

Reviewer #3: Dear author,

Thank you for you submission.

There are some points to consider:

1. The title of the article says that the DED was diagnosed during the COVID-19 pandemic, and the discussion mention that during the pandemic medical students were prone to more stress (by T-PSS-10) and VDT time. There weren’t any comparisons of these variables before and during the pandemic. Moreover, do you have any evidence that the DED symptoms / signs were more severe during the pandemic in comparison to the time before?

2. The article concludes that the medical students in Chiang may had a higher rate of DED comparing other studies. You chose the COVID-19 pandemic as one possible reason. Besides the previous note, some other genetic or environmental factors should be considered.

3. The prevalence of DED was based on OSDI score of >12 and the prevalence was 70.8% which is higher than other articles. However, the articles that were cited used other methods or in combination with OSDI score for diagnosing DED. Therefore it should be mentioned as a limitation.

4. In the discussion you mention female sex was a risk factor for DED, but in the regression analysis (table 3) it was not significant.

5. Another limitation that should be mentioned is that only one observer assessed the Meibomian glands. Did the observer assessed every eyelid separately? Or did he/she assessed all 4 eyelids as one unit knowing that they belong to the same patient? The second option could lead to an observer bias.

6. PLOS authors have the option to publish the peer review history of their article (what does this mean?). If published, this will include your full peer review and any attached files.

Reviewer #1: No

Reviewer #2: No

Reviewer #3: No

---

## [Author Response · Author response to Decision Letter 0]

18 Jul 2021

Dear Editor in Chief and all reviewers

Journal of PLoS ONE

Revision of submitted manuscript: [PONE-D-21-18381] - [EMID:7fa3bce5a1aded7e]

Prevalence of symptomatic dry eye disease with associated risk factors among medical students at Chiang Mai University during COVID-19 pandemic

We do appreciate your valuable time reviewing our manuscript for publication with revision suggestions. So, we would like to submit our thoroughly point-by-point response (blue fonts) to your reviews (black fonts), as stated in the letter of Jun 26, 2021, as follows:

Part I: Journal Requirements:

-The manuscript format is reviewed and meets PLOS ONE's style requirements.

"This research was funded by Faculty of Medicine, Chiang Mai University, Chiang Mai, Thailand

(Grant number: 043/2564). The funder has no role in the study design, data collection and analysis, decision to publish, or preparation of the manuscript."

"This research was funded by Faculty of Medicine, Chiang Mai University, Chiang Mai, Thailand (Grant number: 043/2564). The funder has no role in the study design, data collection and analysis, decision to publish, or preparation of the manuscript. "

- The funding information has been removed from the Acknowledgments section. There has been no change with the current Funding Statement. Please kindly change the online submission form on our behalf. 

 "This research was funded by Faculty of Medicine, Chiang Mai University, Chiang Mai, Thailand (Grant number: 043/2564). The funder has no role in the study design, data collection and analysis, decision to publish, or preparation of the manuscript. "

3. In your Data Availability statement, you have not specified where the minimal data set underlying the results described in your manuscript can be found. PLOS defines a study's minimal data set as the underlying data used to reach the conclusions drawn in the manuscript and any additional data required to replicate the reported study findings in their entirety. All PLOS journals require that the minimal data set be made fully available. 

"Upon re-submitting your revised manuscript, please upload your study’s minimal underlying data set as either Supporting Information files or to a stable, public repository and include the relevant URLs, DOIs, or accession numbers within your revised cover letter. 

Important: If there are ethical or legal restrictions to sharing your data publicly, please explain these restrictions in detail. 

- The information of Availability of Data and Materials has been added as follows: 

 The datasets used and/or analyzed in this study are available from the corresponding author on reasonable request.

4. We note that Figures 1 and 2 in your submission contain copyrighted images. All PLOS content is published under the Creative Commons Attribution License (CC BY 4.0), which means that the manuscript, images, and Supporting Information files will be freely available online, and any third party is permitted to access, download, copy, distribute, and use these materials in any way, even commercially, with proper attribution. 

a. You may seek permission from the original copyright holder of Figures 1 and 2 to publish the content specifically under the CC BY 4.0 license. 

We recommend that you contact the original copyright holder with the Content Permission Form. Please be aware that this license allows unrestricted use and distribution, even commercially, by third parties. Please reply and provide explicit written permission to publish XXX under a CC BY license and complete the attached form.”

- All images in Figures 1 and 2 were obtained from participants in this study with informed consent. Moreover, during revision, we have added more photographic techniques (Reflected IR, Trans IR) with references. (21 and 22) 

21. Adil MY, Xiao J, Olafsson J, Chen X, Lagali NS, Ræder S, et al. Meibomian Gland Morphology Is a Sensitive Early Indicator of Meibomian Gland Dysfunction. Am J Ophthalmol. 2019;200:16-25. Epub 2018/12/24. doi: 10.1016/j.ajo.2018.12.006. PubMed PMID: 30578784.

22. Halleran C, Kwan J, Hom M, Harthan J, editors. Agreement in reading centre grading of meibomian gland tortuosity and atrophy. Poster presented at American Academy of Optometry annual meeting: November; 2016.

And, the additional figure legends for Figure 1 and 2 as follows:

Fig 1. The Meibograde grading system: subjective grading of meibomian gland loss. Row A: Dynamic illumination mode (Reflected infrared) of upper lids; Row B: Dynamic illumination mode (Reflected infrared) of lower lids; Row C: Adaptive transillumination mode (Trans infrared) of lower lids.

Fig 2. The Meibomian gland tortuosity grading system. Row A: Dynamic illumination mode (Reflected infrared) of upper lids; Row B: Dynamic illumination mode (Reflected infrared) of lower lids; Row C: Adaptive transillumination mode (Trans infrared) of lower lids; Row D: Dual mode (combined dynamic illumination and adaptive transillumination) of lower lids.

Part II: Reviewers' comments:

 Reviewer #1: Chulaluck et al. describe a prospective cross-sectional study aimed to explore the prevalence of symptomatic Dry eye disease (DED) with associated risk factors among medical students at Chiang Mai University during covid-19 pandemic

Comments:

1. Abstract: The study did not include any clinical evaluation of the participants; therefore some of them could potentially have other ocular conditions that may have caused dry eye symptoms.

- The study focused on the dry eye symptoms evaluating by the questionnaires and the meibomian gland function evaluating by the LipiView® Interferometer. 

2. The use of a surgical face mask – which is known to increase dry eye symptoms especially during the COVID-19 pandemic, was not part of the risk factors taken in consideration.

- Thank you for your comment. All students are always wearing a face mask during social activities. Therefore, this factor was equally distributed between groups and has not been included in the analysis. 

3. The information collected during the study only represented the participants' condition at the day of their examination.

- Thought the data was collected on one day. However, the questionnaires represented general symptoms during a recent period; moreover, the meibomian gland function does not alter rapidly without any medical interventions. So, our data could effectively describe the prevalence of symptomatic dry eye disease with associated risk factors among medical students at Chiang Mai University during COVID-19 pandemic.

4. Methods section: was not elaborate enough. Exclusion and inclusion criteria are not detailed enough.

- More details of exclusion and inclusion criteria were added in the materials and methods section as follows:

 “Only medical students who completed the questionnaire and the LipiView® II interferometer examination were included in the analysis. All participants can refuse to be in the study at all, or to stop participating at any time of the study.”

And

 “Participants who had a history of ocular surgery or trauma within 3 months, or ocular diseases such as ocular infection, allergy, autoimmune disease, and those using punctal plug or topical ocular medications other than artificial tears were excluded.”

5. The LipiView test is designed to measure the thickness of the lipid layer, but recent studies showed that It doesn’t seem to correlate too well with people’s slit lamp exams or symptoms.

- In our study, lipid thickness was considered as one factor derived from meibomian gland evaluation. However, in the univariate analysis, the lipid thickness was not significantly different among groups of participants with and without dry eye symptoms.

6. The study only included medical students from Chiang Mai University, which may not represent all medical students in Thailand.

- Thank you for your comment. Though our study included medical students from a single university in Thailand, we believe that our data was useful and could represent the general characteristics of dry eye symptoms of medical students in Thailand for two main reasons. First, our study included a large number of medical students (N = 528). Secondly, we believe that the characteristics of medical students in different parts of Thailand should not be markedly different. 

Reviewer #2: We would like to thank the authors for their work on the Prevalence of symptomatic Dry eye disease with associated risk factors among medical students at Chiang Mai University during COVID-19 pandemic. We think it is a well written paper.

We still would like to give some advice on correction:

1.) Please correct the punctuations. There are lots of mistakes regarding the commas options

- The manuscript was revised by a native American speaker. However, we are willing to revise the manuscript again once all corrections are done regarding the English correction. 

Introduction:

2.) Please change “The prevalence of dry eye disease (DED) in Asians is higher than Caucasians.” To “The prevalence of dry eye disease (DED) in Asians is higher than in Caucasians. “ and name a reference

- Thank you for your comment. The correction has been done. However, the references for this statement were of number 5 and 6 as follows:

5. Stapleton F, Alves M, Bunya VY, Jalbert I, Lekhanont K, Malet F, et al. TFOS DEWS II Epidemiology Report. The ocular surface. 2017;15(3):334-65. Epub 2017/07/25. doi: 10.1016/j.jtos.2017.05.003. PubMed PMID: 28736337.

6. Lekhanont K, Rojanaporn D, Chuck RS, Vongthongsri A. Prevalence of dry eye in Bangkok, Thailand. Cornea. 2006;25(10):1162-7. Epub 2006/12/19. doi: 10.1097/01.ico.0000244875.92879.1a. PubMed PMID: 17172891.

3.) “The international workshop defined MGD as a chronic, diffuse abnormality of the meibomian glands, usually characterized by an obstruction of terminal duct and/or changes in the secretion of the glands qualitatively/quantitatively. Please name a reference

- Thank you for your comment. The reference for this definition was in number 7 as follow:

7. Nelson JD, Shimazaki J, Benitez-del-Castillo JM, Craig JP, McCulley JP, Den S, et al. The international workshop on meibomian gland dysfunction: report of the definition and classification subcommittee. Invest Ophthalmol Vis Sci. 2011;52(4):1930-7. Epub 2011/04/01. doi: 10.1167/iovs.10-6997b. PubMed PMID: 21450914; PubMed Central PMCID: PMCPMC3072158.

And the reference has already been mentioned in the following paragraph. 

“The international workshop defined MGD as a chronic, diffuse abnormality of the meibomian glands, usually characterized by an obstruction of terminal duct and/or changes in the secretion of the glands qualitatively/quantitatively. The condition may lead to a tear film alteration, symptoms of eye irritation and inflammation, and ocular surface disease [7].”

4.) The survey questionnaire had four sections: demographic data with pre-existing medical conditions; risk factors for DED including personal habits; psychological stress; and a dry eye questionnaire using the Ocular Surface Disease Index (OSDI), please attach the questionnaire

- Thank you for your comment. The questionnaire will be added as a supplemental in both Thai and English versions. 

5.) Please change “Participants who used artificial tears were instructed to stop using for 6 hours before LipiView® II examination. “ to “Participants who used artificial tears were instructed to stop the use 6 hours before LipiView® II examination.”

- Thank you for your comment. The change has been done as suggested.

LipiView® II Ocular Surface interferometer:

6.) “For the blinking pattern, the incomplete blinking ratio was calculated by the number of incomplete blinks divided by the total blinks.” Please refer to the attached picture

- Blinking pattern is used to assess the quality of blinking. The best quality of blinking pattern is that every blink is complete (defined as the complete apposition of upper and lower lids.). The LipiView could demonstrate the number of incomplete and complete blinks per 20 seconds and also calculate the partial blinking rate (PBR). The reference 20 is added. However, there was no figure for the blinking pattern.

20. Eom Y, Lee JS, Kang SY, Kim HM, Song JS. Correlation between quantitative measurements of tear film lipid layer thickness and meibomian gland loss in patients with obstructive meibomian gland dysfunction and normal controls. Am J Ophthalmol. 2013;155(6):1104-10.e2. Epub 2013/03/08. doi: 10.1016/j.ajo.2013.01.008. PubMed PMID: 23465270.

Results:

7.) 392/528 is 74,24% not 85%

- Thank you for your comment. The change has been done as suggested. (in Results)

8.) 69/528 is 13% not 15%

- Thank you for your comment. The change has been done as suggested. (in Results)

Table 1:

9.)

10.) Please change “average” to “average”

- The change has been done as suggested. (in Table 1)

Discussion:

11.) “Hyon et al.“ Please also name the reference here

- Thank you for your comment. The reference for this definition was in number 15 as follow:

15. Hyon JY, Yang HK, Han SB. Dry Eye Symptoms May Have Association With Psychological Stress in Medical Students. Eye & contact lens. 2019;45(5):310-4. Epub 2018/12/27. doi: 10.1097/icl.0000000000000567. PubMed PMID: 30585856.

And the reference has already been mentioned in the following paragraph. 

“Hyon et al. demonstrated that stress, female sex, contact lens wear, and duration of using VDT were significant risk factors for DED. The prevalence of DED based on the symptoms in Korean medical students was 27.1% [15].”

12.) “Health promotion with the “20-20-20” rule and limited screen time may be very helpful.“ Please explain this rule

- Thank you for your comment. The explanation has been added as suggested.

“Health promotion with the “20-20-20” rule, which recommends that every 20 minutes, an individual should take a 20-second break and focus their eyes on something at least 20 feet away. and limited screen time may be very helpful.”

Reviewer #3: Dear author,

Thank you for you submission.

There are some points to consider:

1. The title of the article says that the DED was diagnosed during the COVID-19 pandemic, and the discussion mention that during the pandemic medical students were prone to more stress (by T-PSS-10) and VDT time. There weren’t any comparisons of these variables before and during the pandemic. Moreover, do you have any evidence that the DED symptoms / signs were more severe during the pandemic in comparison to the time before?

- Thank you for your comment. As the study was conducted for the first time during the pandemic, there has been no previous report on the prevalence of dry eye symptoms in this specific population. However, regarding the previous study of the prevalence of dry eye in Thai adults which reported to be 34% [5], the prevalence of dry eye in our study was higher.

2. The article concludes that the medical students in Chiang may had a higher rate of DED comparing other studies. You chose the COVID-19 pandemic as one possible reason. Besides the previous note, some other genetic or environmental factors should be considered.

- Thank you for your comment. Please allow us to clarify the COVID-19 pandemic as a possible reason of higher prevalence of dry eye symptoms. In our study, the COVID-19 pandemic was taken into consideration as the pandemic leads to both behavioral and environmental changes that may aggravate the dry eye symptoms. For example, during the pandemic, most of lectures were conducted via the online programmes which lead to more screen time. Also, the circumstances may stress all the medical students in many ways as discussed in the manuscript. In our study, the genetic factors, though might be significant, but was not included. 

3. The prevalence of DED was based on OSDI score of >12 and the prevalence was 70.8% which is higher than other articles. However, the articles that were cited used other methods or in combination with OSDI score for diagnosing DED. Therefore it should be mentioned as a limitation.

- Thank you for your comment. The following statement has been added to the limitation part.

“Moreover, the prevalence of DED may be varied from the different diagnostic methods.”

4. In the discussion you mention female sex was a risk factor for DED, but in the regression analysis (table 3) it was not significant.

- Thank you for your comment. That is correct that in the regression analysis, the p value for female sex was 0.052. Therefore, we stated that female sex was likely to be associated with an increased risk of DED, similar to the previous studies.

5. Another limitation that should be mentioned is that only one observer assessed the Meibomian glands. Did the observer assessed every eyelid separately? Or did he/she assessed all 4 eyelids as one unit knowing that they belong to the same patient? The second option could lead to an observer bias.

- Thank you for your comment. In this study, a single experienced observer (CT) subjectively evaluated the meibomian gland dropouts in both upper and lower eyelids using a validated Meibograde grading scheme. Each eyelid was blinded for evaluation and the upper and lower lids were separately evaluated. 

We appreciate for the very helpful review.

Sincerely Yours,

Chulaluck Tangmonkongvoragul, MD

Department of Ophthalmology, 

Faculty of Medicine, 

Chiang Mai University, 

Chiang Mai, Thailand 

E-mail: poupae025@gmail.com, chulaluck.t@cmu.ac.th

---

## [Decision Letter · Decision Letter 1]

23 Aug 2021

PONE-D-21-18381R1

Prevalence of symptomatic dry eye disease with associated risk factors among medical students at Chiang Mai University during COVID-19 pandemic

PLOS ONE

Dear Dr. Tangmonkongvoragul,

Thank you for submitting your manuscript to PLOS ONE. After careful consideration, we feel that it has merit but does not fully meet PLOS ONE’s publication criteria as it currently stands. Therefore, we invite you to submit a revised version of the manuscript that addresses the points raised during the review process.

We look forward to receiving your revised manuscript.

Kind regards,

Michael Mimouni

Academic Editor

PLOS ONE

Journal Requirements:

Reviewers' comments:

Reviewer's Responses to Questions

**Comments to the Author**

1. If the authors have adequately addressed your comments raised in a previous round of review and you feel that this manuscript is now acceptable for publication, you may indicate that here to bypass the “Comments to the Author” section, enter your conflict of interest statement in the “Confidential to Editor” section, and submit your "Accept" recommendation.

Reviewer #1: All comments have been addressed

Reviewer #2: All comments have been addressed

Reviewer #3: All comments have been addressed

2. Is the manuscript technically sound, and do the data support the conclusions?

Reviewer #1: Partly

Reviewer #2: Yes

Reviewer #3: Yes

3. Has the statistical analysis been performed appropriately and rigorously? 

Reviewer #1: Yes

Reviewer #2: Yes

Reviewer #3: N/A

4. Have the authors made all data underlying the findings in their manuscript fully available?

Reviewer #1: Yes

Reviewer #2: Yes

Reviewer #3: No

5. Is the manuscript presented in an intelligible fashion and written in standard English?

Reviewer #1: Yes

Reviewer #2: Yes

Reviewer #3: Yes

6. Review Comments to the Author

Reviewer #1: Thank you for addressing my comments, you did provided reasonable explanations. The article does provide interesting information regarding the influence of the COVID19 pandemic on medical students in Chiang Mai University.

I would recommend it to be published.

Reviewer #2: We would like to thank the authors for editing their work. We do not have any other points to edit.

Reviewer #3: Dear author,

Thank you for your response.

There are still issues that need to be addressed correctly:

1. Explaining the prevalence of DED by “the pandemic” is not correct, you should evaluate each factor separately. Please add references for comparison of the following factors before and after the pandemic – dry eye prevalence in similar age group (comparing to adults is not sufficient); students’ psychological stress; students’ computer and screen time use.

2. If every eyelid was evaluated separately by the blinded single observer, this should be noted.

7. PLOS authors have the option to publish the peer review history of their article (what does this mean?). If published, this will include your full peer review and any attached files.

Reviewer #1: No

Reviewer #2: No

Reviewer #3: No

---

## [Author Response · Author response to Decision Letter 1]

26 Aug 2021

Dear Editor in Chief and all reviewers

Journal of PLoS ONE

Revision of submitted manuscript: [PONE-D-21-18381R1] 

Prevalence of symptomatic dry eye disease with associated risk factors among medical students at Chiang Mai University during COVID-19 pandemic

We do appreciate your valuable time reviewing our manuscript for publication with revision suggestions. So, we would like to submit our thoroughly point-by-point response to your reviews, as stated in the letter of Aug 23, 2021, as follows:

6. Review Comments to the Author

Reviewer #3: Dear author, Thank you for your response. There are still issues that need to be addressed correctly:

1. Explaining the prevalence of DED by “the pandemic” is not correct, you should evaluate each factor separately. Please add references for comparison of the following factors before and after the pandemic – dry eye prevalence in similar age group (comparing to adults is not sufficient); students’ psychological stress; students’ computer and screen time use.

- Thank you for your comment. In our study, the pandemic is not directly a risk factor for DED in medical students, but rather be a unique situation that interfered with the known risk factors for DED. We believe that during the pandemic, medical students had to dramatically change the ways of learning and also their lifestyles which might affect with the DED symptoms. However, due to limited number of the DED prevalence in a specific population like medical students, the comparison group as general adult population might be sufficient, and further studies of the DED prevalence in medical students in Thailand when the pandemic is subsided, is an interesting idea for our future project.

2. If every eyelid was evaluated separately by the blinded single observer, this should be noted.

-Thank you for your comment. The following statement has been added to the methods part.

A single experienced observer (CT) subjectively evaluated the meibomian gland dropout in both upper and lower eyelids using a validated Meibograde grading scheme, with a 4-point scale from 0 to 3 in which grade 0 is 0-25% meibomian gland loss; grade 1, 26-50% loss; grade 2, 51-75% loss; and grade 3, more than 75% loss [21]. The meibomian gland loss was calculated with reference to the equivalent meibomian gland area in healthy individuals. Meibomian gland tortuosity for each eyelid was graded using the 5-point Halleran scale: grade 0, no tortuosity; grade 1, less than 25% tortuosity; grade 2, 26-50% tortuosity; grade 3, 51-74% tortuosity; and grade 4, more than 75% tortuosity [22]. Each eyelid was blinded for evaluation and the upper and lower lids were separately evaluated. 

We appreciate for the very helpful review.

Sincerely Yours,

Chulaluck Tangmonkongvoragul, MD

Department of Ophthalmology, 

Faculty of Medicine, 

Chiang Mai University, 

Chiang Mai, Thailand 

E-mail: poupae025@gmail.com, chulaluck.t@cmu.ac.th

---

## [Decision Letter · Decision Letter 2]

15 Nov 2021

PONE-D-21-18381R2Prevalence of symptomatic dry eye disease with associated risk factors among medical students at Chiang Mai University during COVID-19 pandemicPLOS ONE

Dear Dr. Tangmonkongvoragul,

Thank you for submitting your manuscript to PLOS ONE. After careful consideration, we feel that it has merit but does not fully meet PLOS ONE’s publication criteria as it currently stands. Therefore, we invite you to submit a revised version of the manuscript that addresses the points raised during the review process.

We look forward to receiving your revised manuscript.

Kind regards,

Michael Mimouni

Academic Editor

PLOS ONE

Journal Requirements:

Reviewers' comments:

Reviewer's Responses to Questions

**Comments to the Author**

1. If the authors have adequately addressed your comments raised in a previous round of review and you feel that this manuscript is now acceptable for publication, you may indicate that here to bypass the “Comments to the Author” section, enter your conflict of interest statement in the “Confidential to Editor” section, and submit your "Accept" recommendation.

Reviewer #2: (No Response)

Reviewer #3: All comments have been addressed

2. Is the manuscript technically sound, and do the data support the conclusions?

Reviewer #2: Yes

Reviewer #3: Yes

3. Has the statistical analysis been performed appropriately and rigorously? 

Reviewer #2: Yes

Reviewer #3: N/A

4. Have the authors made all data underlying the findings in their manuscript fully available?

Reviewer #2: Yes

Reviewer #3: Yes

5. Is the manuscript presented in an intelligible fashion and written in standard English?

Reviewer #2: Yes

Reviewer #3: Yes

6. Review Comments to the Author

Reviewer #2: We do thank the authors for addressing our comments. We still have one point to address. We think having an adequate control group is essential. Please look for better comparison to make your data valide. Please look for DED in a more comparable group like other students before the pandemic.

Compare the risk factors for DED one by one before and after the pandemic like "psychological stress", screen and computer time.

Reviewer #3: (No Response)

7. PLOS authors have the option to publish the peer review history of their article (what does this mean?). If published, this will include your full peer review and any attached files.

Reviewer #2: No

Reviewer #3: No

---

## [Author Response · Author response to Decision Letter 2]

10 Dec 2021

Dear Editor in Chief and all reviewers

Journal of PLoS ONE

Revision of submitted manuscript: [PONE-D-21-18381R2] 

Prevalence of symptomatic dry eye disease with associated risk factors among medical students at Chiang Mai University during COVID-19 pandemic

We do appreciate your valuable time reviewing our manuscript for publication with revision suggestions. So, we would like to submit our thoroughly point-by-point response (blue fonts) to your reviews (black fonts), as stated in the letter of Nov 16, 2021, as follows:

6. Review Comments to the Author

Reviewer #2: We do thank the authors for addressing our comments. We still have one point to address. We think having an adequate control group is essential. Please look for better comparison to make your data valid. Please look for DED in a more comparable group like other students before the pandemic.

Compare the risk factors for DED one by one before and after the pandemic like "psychological stress", screen and computer time.

- Thank you for your comment. We have added the findings of Hyon et al. study which conducted the study of DED in Asian medical students before the COVID-19 pandemics in the second paragraph of the discussion part as follows:

Hyon et al. demonstrated that stress, female sex, contact lens wear, and duration of using VDT were significant risk factors for DED. The prevalence of DED based on the symptoms in Korean medical students was 27.1%. This study evaluated DED in medical students before the COVID-19 pandemics and revealed that DED may have association with psychological stress (using the Perceived Stress Scale 4 (PSS-4) questionnaire) [15]. Additionally, the PSS scores in our study tended to be higher than the study by Hyon et al., though the direct comparison was not allowed due to the different versions used. Therefore, a higher prevalence of symptomatic DED in medical students (70.8%) in our study may be explained by more stressful situation during the COVID-19 pandemic among medical students.

We appreciate for the very helpful review.

Sincerely Yours,

Chulaluck Tangmonkongvoragul, MD

Department of Ophthalmology, 

Faculty of Medicine, 

Chiang Mai University, 

Chiang Mai, Thailand 

E-mail: poupae025@gmail.com, chulaluck.t@cmu.ac.th

---

## [Decision Letter · Decision Letter 3]

24 Jan 2022

PONE-D-21-18381R3Prevalence of symptomatic dry eye disease with associated risk factors among medical students at Chiang Mai University during COVID-19 pandemicPLOS ONE

Dear Dr. Tangmonkongvoragul,

Thank you for submitting your manuscript to PLOS ONE. After careful consideration, we feel that it has merit but does not fully meet PLOS ONE’s publication criteria as it currently stands. Therefore, we invite you to submit a revised version of the manuscript that addresses the points raised during the review process.

We look forward to receiving your revised manuscript.

Kind regards,

Michael Mimouni

Academic Editor

PLOS ONE

Journal Requirements:

Reviewers' comments:

Reviewer's Responses to Questions

**Comments to the Author**

1. If the authors have adequately addressed your comments raised in a previous round of review and you feel that this manuscript is now acceptable for publication, you may indicate that here to bypass the “Comments to the Author” section, enter your conflict of interest statement in the “Confidential to Editor” section, and submit your "Accept" recommendation.

Reviewer #2: All comments have been addressed

Reviewer #3: All comments have been addressed

2. Is the manuscript technically sound, and do the data support the conclusions?

Reviewer #2: Yes

Reviewer #3: Yes

3. Has the statistical analysis been performed appropriately and rigorously? 

Reviewer #2: Yes

Reviewer #3: N/A

4. Have the authors made all data underlying the findings in their manuscript fully available?

Reviewer #2: Yes

Reviewer #3: Yes

5. Is the manuscript presented in an intelligible fashion and written in standard English?

Reviewer #2: Yes

Reviewer #3: Yes

6. Review Comments to the Author

Reviewer #2: We would like to thank the authors for their work on the Prevalance of symptomatic Dry Eye Disease with associated risk factors among medical students at Chiang Mai University during the COVID19 pandemic. We think it is a well written paper. All comments have been adressed.

Reviewer #3: Thank you for you submission.

Because it may be somehow misleading, i recommend the editor to reconsider the title of the article "during COVID-19 pandemic" as there are no direct causes for dry eye due to COVID-19 as the manuscirpt suggest, but only indirect risk factors (screen time, face masks).

all other notes has been adressed

7. PLOS authors have the option to publish the peer review history of their article (what does this mean?). If published, this will include your full peer review and any attached files.

Reviewer #2: No

Reviewer #3: No

---

## [Author Response · Author response to Decision Letter 3]

7 Feb 2022

Dear Editor in Chief and all reviewers

Journal of PLOS ONE

Revision of submitted manuscript: [PONE-D-21-18381R3] 

Prevalence of symptomatic dry eye disease with associated risk factors among medical students at Chiang Mai University during COVID-19 pandemic

We do appreciate your valuable time reviewing our manuscript for publication with revision suggestions. So, we would like to submit our thoroughly point-by-point response to your reviews, as stated in the letter of Jan 24, 2022, as follows:

6. Review Comments to the Author

Reviewer #3: Thank you for you submission.

Because it may be somehow misleading, I recommend the editor to reconsider the title of the article "during COVID-19 pandemic" as there are no direct causes for dry eye due to COVID-19 as the manuscript suggest, but only indirect risk factors (screen time, face masks).

all other notes has been addressed.

- Thank you for your comment. We have changed the title of the article to avoid misleading as follows: 

Prevalence of symptomatic dry eye disease with associated risk factors among medical students at Chiang Mai University due to increased screen time and stress during COVID-19 pandemic

We appreciate for the very helpful review.

Sincerely Yours,

Chulaluck Tangmonkongvoragul, MD

Department of Ophthalmology, 

Faculty of Medicine, 

Chiang Mai University, 

Chiang Mai, Thailand 

E-mail: poupae025@gmail.com, chulaluck.t@cmu.ac.th

---

## [Decision Letter · Decision Letter 4]

8 Mar 2022

Prevalence of symptomatic dry eye disease with associated risk factors among medical students at Chiang Mai University due to increased screen time and stress during COVID-19 pandemic

PONE-D-21-18381R4

Dear Dr. Tangmonkongvoragul,

We’re pleased to inform you that your manuscript has been judged scientifically suitable for publication and will be formally accepted for publication once it meets all outstanding technical requirements.

Kind regards,

Michael Mimouni

Academic Editor

PLOS ONE

Additional Editor Comments (optional):

Reviewers' comments:

Reviewer's Responses to Questions

**Comments to the Author**

1. If the authors have adequately addressed your comments raised in a previous round of review and you feel that this manuscript is now acceptable for publication, you may indicate that here to bypass the “Comments to the Author” section, enter your conflict of interest statement in the “Confidential to Editor” section, and submit your "Accept" recommendation.

Reviewer #2: All comments have been addressed

Reviewer #3: All comments have been addressed

2. Is the manuscript technically sound, and do the data support the conclusions?

Reviewer #2: Yes

Reviewer #3: Yes

3. Has the statistical analysis been performed appropriately and rigorously? 

Reviewer #2: Yes

Reviewer #3: I Don't Know

4. Have the authors made all data underlying the findings in their manuscript fully available?

Reviewer #2: Yes

Reviewer #3: Yes

5. Is the manuscript presented in an intelligible fashion and written in standard English?

Reviewer #2: Yes

Reviewer #3: Yes

6. Review Comments to the Author

Reviewer #2: (No Response)

Reviewer #3: (No Response)

7. PLOS authors have the option to publish the peer review history of their article (what does this mean?). If published, this will include your full peer review and any attached files.

Reviewer #2: No

Reviewer #3: No

---

## [Editor Report · Acceptance letter]

14 Mar 2022

PONE-D-21-18381R4 

Prevalence of symptomatic dry eye disease with associated risk factors among medical students at Chiang Mai University due to increased screen time and stress during COVID-19 pandemic 

Dear Dr. Tangmonkongvoragul:

I'm pleased to inform you that your manuscript has been deemed suitable for publication in PLOS ONE. Congratulations! Your manuscript is now with our production department. 

Kind regards, 

on behalf of

Dr. Michael Mimouni 

Academic Editor

PLOS ONE